# Idempotent Learned Image Compression with Right-Inverse

**Yanghao Li,  Tongda Xu,  Yan Wang**\*  **Jingjing Liu,  Ya-Qin Zhang**\*
Institute for AI Industry Research (AIR), Tsinghua University
liyangha18@mails.tsinghua.edu.cn,    wangyan@air.tsinghua.edu.cn

## Abstract

We consider the problem of idempotent learned image compression (LIC). The idempotence of codec refers to the stability of codec to re-compression. To achieve idempotence, previous codecs adopt invertible transforms such as DCT [Wallace, 1991] and normalizing flow [Papamakarios et al., 2021]. In this paper, we first identify that invertibility of transform is sufficient but not necessary for idempotence. Instead, it can be relaxed into right-invertibility. And such relaxation allows wider family of transforms. Based on this identification, we then implement an idempotent codec using our proposed blocked convolution and null-space enhancement. Empirical results show that we achieve state-of-the-art rate-distortion performance among idempotent codecs. Furthermore, our idempotent codec can be extended into near-idempotent codec by relaxing the right-invertibility. And this near-idempotent codec has significantly less quality decay after 50 rounds of re-compression compared with other near-idempotent codecs.

## 1 Introduction

Learned Image Compression (LIC) has been widely studied in recent years [Ballé et al., 2017, 2018, Minnen et al., 2018, Cheng et al., 2020, Minnen et al., 2020, He et al., 2021, 2022] and has shown promising rate-distortion (RD) performance. However, the loss caused by re-compression is much more severe in LIC compared with traditional codec, which seriously limits the practical application of LIC [Kim et al., 2020]. In this paper, we study the idempotence of LIC, which refers to the stability of codec to re-compression. More specifically, denote the original image as $\boldsymbol{x}$, the encode-then-decode procedure as $f(\cdot)$, and the reconstructed image as $f(\boldsymbol{x})$, we say a codec is idempotent if:

$$f(\boldsymbol{x}) = f(f(\boldsymbol{x})). \tag{1}$$

For traditional codecs such as JPEG [Wallace, 1991] and JPEG2000 [Taubman et al., 2002], idempotence is easily achieved. This is because those codecs adopt invertible transforms such as Discrete Cosine transform (DCT) and Discrete Wavelet transform (DWT). And the only non-invertible operation is the scalar quantization. As scalar quantization using rounding is naturally idempotent, the idempotence of the whole codec can be easily assured. For LIC, however, neural-network-based transform is introduced for expressiveness. And as most neural networks are non-invertible, the idempotence of LIC is not trivial.

A natural solution to this problem is replacing the non-invertible encoding transform with invertible ones. [Helminger et al., 2021] construct the encoding transform with only invertible normalizing flow [Papamakarios et al., 2021]. However, due to the limited expressiveness of invertible operations, a dramatic RD performance drop is observed. Another line of works are targeted at near-idempotence, which means that they achieve a small $|f(x) - f(f(x))|$, but not $f(x) = f(f(x))$. These works

---

\*Corresponding author.

37th Conference on Neural Information Processing Systems (NeurIPS 2023).

adopt partially invertible encoding transform [Cai et al., 2022] or use additional regularization loss [Kim et al., 2020] to constrain re-compression loss, but none of them is able to achieve idempotence like JPEG and JPEG2000.

In this work, we first identify that invertibility is sufficient but not necessary for idempotent codecs. Instead, it can be relaxed into right-invertibility, and such relaxation allows for more flexible and expressive encoding transforms. Based on this observation, we investigate practical implementation of right-inverse, and propose a highly efficient blocked convolution to overcome the forbidding time complexity of right-inverse. Additionally, we leverage the null space decomposition [Schwab et al., 2019, Wang et al., 2022a,b] to further boost the expressiveness of the right-invertible encoding transform. Empirically, we achieve state-of-the-art RD performance among existing idempotent codecs. Further, our idempotent codec can be easily extended into a near-idempotent codec, which also achieves state-of-the-art re-compression performance among near-idempotent codecs.

## 2   Sufficiency of Right-Invertibility for Idempotence

Most modern LIC [Ballé et al., 2017, 2018, Cheng et al., 2020, He et al., 2022] are composed of four components: the encoding transform $E$, the decoding transform $D$, the quantization $Q$, and the entropy model. On the encoder side, the encoding transform $E$ transform the input image $x$ to some latent representation, which is then quantized by $Q$ to give the code $y$. The entropy model models the entropy of code $y$, and is then used losslessly compress $y$ to a bitstream, On the decoder side, the received bitstream is losslessly decompressed to code $y$ with the help of the entropy model. After that, the decoding transform $D$ transforms the $y$ to the decompressed $\hat{x}$. This compress-decompress procedure can be represented as

$$y = Q \circ E(x), \ \hat{x} = D(y). \tag{2}$$

Note that the code $y$ is losslessly encoded and decoded, regardless of how good the entropy model is. Therefore entropy model does not influence the distortion, and is thus omitted for simplicity.

Using similar notations, the re-compression cycle can be represented as

$$y_n = Q \circ E(x_{n-1}), \ x_n = D(y_n) = D \circ Q \circ E(x_{n-1}), \tag{3}$$

where $x_0$ is the original image, $y_n$ is the code after $n$ times' re-compression, and $x_n$ is the corresponding decompressed image. We say a codec is idempotent if and only if

$$x_n = x_1 = D \circ Q \circ E(x_0), \forall n \geq 1. \tag{4}$$

An obvious sufficient condition for idempotence (or for Eq. 4) is

$$D = E^{-1}. \tag{5}$$

In other words, $D$ and $E$ are inverse of each other. Then, the only difference between $x_1$ and $x_n$ is the number of applications of quantization operation $Q$:

$$x_n = D \circ Q \circ \cancel{E \circ D} \circ Q \circ E(x_{n-2}) = ... = D \circ Q^n \circ E(x_0). \tag{6}$$

This re-compression procedure is idempotent as long as $Q$ is idempotent. Usually $Q$ is a scalar quantization implemented by rounding, then this procedure is naturally idempotent. The invertibility of encoding transform is satisfied in traditional image codecs like JPEG and JPEG2000, and can also be achieved using normalizing flow [Helminger et al., 2021], thus these codecs are idempotent. However, invertibility of $E$ (Eq. 5) is sufficient but not necessary. We notice that for $E \circ D$ in Eq. 6 to be canceled out, it is sufficient to have

$$E \circ D(y_i) = y_i, \tag{7}$$

which is exactly the definition of right-invertibility.

Relaxing the requirement for $E$ from invertibility to right-invertibility brings two advantages. On the one hand, the family of right-invertible functions is much less constrained than the family of invertible functions, which means we have wider choices for $E$ and its easier to improve the expressiveness of $E$. On the other hand, the sufficient condition for $E$ to be invertible is that $E$ is bijective, and the sufficient condition for $E$ to be right-invertible is that $E$ is surjective [Mac Lane, 2013]. A surjective $E$ can transform different input images $x$ into the same code to save the bits for distinguishing them, while bijective $E$ always transforms different input images into different codes. As later shown in the experiment section 4.3, this bit saving property benefits the RD performance of lossy compression.

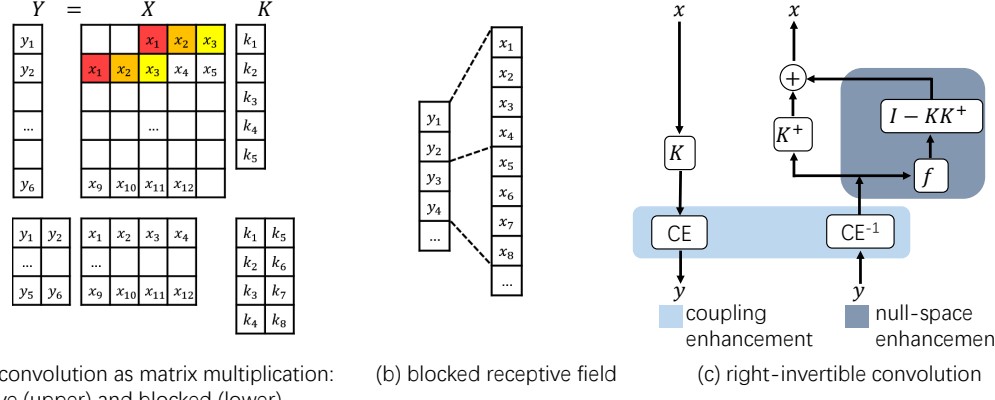

(a) convolution as matrix multiplication: naive (upper) and blocked (lower)

(b) blocked receptive field

(c) right-invertible convolution

Figure 1: Right-Invertible Convolution. (a) Naïve convolution matrix (upper) and blocked convolution matrix (lower). (b) Receptive field pattern for the proposed blocked convolution. (c) Null-space enhancement (NE) and coupling enhancement (CE) to improve expressiveness.

## 3 Practical Design of Right-Invertible Codec

In the previous section, we have shown that if an encoding transform $E$ and quantization $Q$ are right-invertible, we only need the decoding transform $D$ to be the right-inverse of $E$ to formulate an idempotent codec. Since the success of LIC is, to a large extent, owing to the expressive power of the learned encoding and decoding transforms, our task becomes how to design an expressive yet right-invertible encoding transform. As composition of surjections is still surjective [Mac Lane, 2013], this task can be further decomposed into designing small components of right-invertible transforms and combining them altogether. Specifically, we construct the encoding transform $E$ in a composition form as $E = E_n \circ E_{n-1} \circ ... \circ E_1$, and make each $E_i$ a right-invertible sub-transform.

This section is organised as follows: In Sec. 3.1-Sec.. 3.3, we discuss how to design expressive yet right-invertible atom transforms used in LIC, such as convolution, normalization and quantization. In Sec. 3.4, we discuss how to organize those atom transforms into an idempotent codec. And In Sec. 3.5, we discuss how to relax this idempotent codec into near-idempotent codec.

### 3.1 Efficient & Expressive Right-Invertible Convolution

Convolution is of great importance to LIC and makes up the majority of computation cost. Here, we discuss how to implement right-invertible convolution with efficiency and expressiveness. The overall design of right-invertible convolution is illustrated in Fig.1(c).

#### 3.1.1 Blocked Convolution for Efficiency

The right-inverse of a convolution can be calculated in serial (if it exists), but the time complexity is forbiddingly high. To see why, consider a 1-d convolution with kernel size 5, padding 2, stride 2, and channel 1. The input and output are $\boldsymbol{x} = (x_1, x_2, ..., x_{12})$ and $\boldsymbol{y} = (y_1, y_2, ..., y_6)$ respectively.

The serial solution of right-inverse goes as follows: first solve $(x_1, x_2, x_3)$ given $y_1$, then solve $(x_4, x_5)$ given $y_2$ and *already solved* $(x_1, x_2, x_3)$, and so on till the whole $\boldsymbol{x}$ is solved. This serial solution cannot be made parallel because solving $(x_4, x_5)$ needs $(x_1, x_2, x_3)$ to be *already solved*, and is thus extremely time-consuming. The fundamental reason for the dependency in solving $\boldsymbol{x}$ is that, some *same* $x_i$ is involved in the forward calculation of *different* $y_i$, i.e., the overlapping receptive field. Therefore, if we make the receptive field non-overlapping, then parallel solution of right-inverse becomes possible.

Inspired by the non-overlapping $1 \times 1$ convolution for inverse [Kingma and Dhariwal, 2018], we propose blocked convolution for right-inverse. Blocked convolution is also non-overlapping, but extends invertibility to right-invertibility. As shown in Sec. 4.3, this extension boots the the R-D performance of idempotent LIC by a large margin.

With this non-overlapping blocked convolution, we can make solution of right-inverse parallel. Following the previous example, using the same input and output with a $4 \times 2$ blocked-convolution kernel, for now, solving $(x_1, ..., x_4)$ only needs to know $(y_1, y_2)$, and solving $(x_5, ..., x_8)$ only needs to know $(y_3, y_4)$, and these procedures can be made parallel. Matrix multiplication equivalents of normal convolution and blocked convolution are depicted in the upper and lower parts of Fig.1(a) respectively, using the well-known GEMM [GEM] formation. And an analysis of time complexity of 2D convolution can be found in appendix A.1.

### 3.1.2 Null-Space Enhancement and Coupling Enhancement for Expressiveness

**Null-Space Enhancement**

To solve the right-inverse of the proposed blocked convolution in parallel, we adopt the widely-used Moore–Penrose pseudo-inverse [Moo] as

$$X = YK^+, \tag{8}$$

where $Y \in \mathbb{R}^{b \times d}$ is the output of blocked convolution, $X \in \mathbb{R}^{b \times D}$ is the solved right-inverse, and $K^+ \in \mathbb{R}^{d \times D}$ is the Moore-Penrose pseudo-inverse of the kernel $K \in \mathbb{R}^{D \times d}$. $b$ is the batch size, $D$ and $d$ are the input and output dimension of convolution in GEMM formation [GEM].

However, simple Moore-Penrose pseudo-inverse does not have enough expressiveness. This is because right-inverse of a surjection might not be unique [Mac Lane, 2013], and the Moore-Penrose pseudo-inverse might not be the optimal choice to calculate the right-inverse for blocked convolution. Therefore, we leverage the null-space decomposition [Schwab et al., 2019, Wang et al., 2022a,b] to identify a right-inverse that is most suitable. More specifically, Eq. 8 can be extended to

$$X = YK^+ + F(I - KK^+). \tag{9}$$

Here $F \in \mathbb{R}^{b \times D}$ can be an arbitrarily chosen variable. Utilizing this property, we propose to learn a function $f(Y)$ to identify an $F$ that is the most suitable as:

$$X = YK^+ + f(Y)(I - KK^+). \tag{10}$$

We call this approach null-space enhancement (NE). In this way, right-inverse $X$ can be made more than just linear transformation of $Y$ and thus become more expressive. Ablation study of the proposed null-space enhancement can be found in Sec.4.3. Derivation of Eq.9 and parameterization of kernel $K$ can be found in appendix A.2.

**Coupling Enhancement**

The proposed blocked convolution makes the calculation of right-inverse parallel-friendly, but it also restricts the receptive field to a blocked pattern (Fig. 1(b)). This restriction limits the exchange of information across different spatial locations. To overcome this drawback, we propose to introduce a coupling enhancement (CE) after the blocked convolution.

Specifically, we implement a coupling structure [Dinh et al., 2016], and utilize normal convolutions without blocked limitation as its scale and translation functions. This structure is formulated as

$$\boldsymbol{x} = [\boldsymbol{x}_1 \ \boldsymbol{x}_2], \ \boldsymbol{y}_1 = \boldsymbol{x}_1, \ \boldsymbol{y}_2 = \boldsymbol{x}_2 \odot \exp\left(s(\boldsymbol{x}_1)\right) + t(\boldsymbol{x}_1), \ \boldsymbol{y} = [\boldsymbol{y}_1 \ \boldsymbol{y}_2] \tag{11}$$

Here, $\boldsymbol{x}$ and $\boldsymbol{y}$ are the input and output, respectively. $[\cdot]$ is the split/concatenate operation along the channel dimension, and $\odot$ is the element-wise multiplication. $s(\cdot)$ and $t(\cdot)$ are the scale and translation functions, respectively. This structure is fully invertible as

$$\boldsymbol{y} = [\boldsymbol{y}_1 \ \boldsymbol{y}_2], \ \boldsymbol{x}_1 = \boldsymbol{y}_1, \ \boldsymbol{x}_2 = (\boldsymbol{y}_2 - t(\boldsymbol{x}_1))/\exp\left(s(\boldsymbol{x}_1)\right), \ \boldsymbol{x} = [\boldsymbol{x}_1 \ \boldsymbol{x}_2] \tag{12}$$

Note that the invertibility of this coupling structure does not require the invertibility of the scale and translation functions $s(\cdot)$ and $t(\cdot)$, thus $s(\cdot)$ and $t(\cdot)$ can be arbitrary learned transforms. Since the proposed coupling enhancement utilize normal convolutions as its $s(\cdot)$ and $t(\cdot)$, its receptive field is not restricted to be blocked, and can thus serve as an enhancement of expressiveness.

### 3.2 Right-Invertible Generalized Divisive Normalization

The widely-used generalized normalization (GDN)[Ballé et al., 2015] in LIC is invertible in theory, and is thus qualified as a surjection. However, the inverse of GDN has to be solved for every input

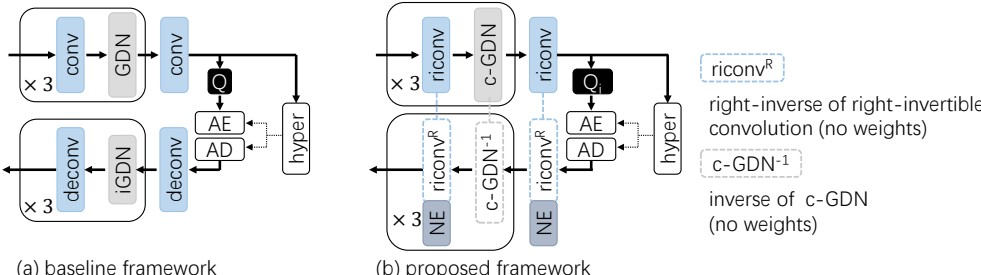

(a) baseline framework     (b) proposed framework

Figure 2: Comparison between (a) baseline framework [Ballé et al., 2018] and (b) proposed framework. To make the framework idempotent, we replace conv/deconv with right-invertible convolution (described in Sec. 3.1), GDN/iGDN with c-GDN (described in Sec. 3.2), and quantization $Q$ with idempotent quantization $Q_i$ (described in Sec. 3.3) Both frameworks use the same mean-scale Gaussian entropy model [Minnen et al., 2018]. AE/AD are arithmetic encoder/decoder, respectively.

image in an iterative manner, and is even not guaranteed to converge in finite steps. Therefore, the original GDN is not suitable for idempotent compression.

We propose a coupling GDN layer (c-GDN) that combines the coupling structure [Dinh et al., 2016] and GDN. The inverse of c-GDN layer can be solved in a much simpler analytical manner. Specifically, we implement a coupling structure [Dinh et al., 2016] with normal GDN as its scale and translation functions ($s(\cdot)$ and $t(\cdot)$). And just like the aforementioned coupling enhancement, the forward and inverse of this c-GDN layer can be calculated according to Eq. 11 and Eq. 12, respectively. We demonstrate empirically in Sec. 4.3 that the proposed c-GDN layer can achieve comparable RD performance with the original GDN.

### 3.3 Right-Invertible Quantization

As previously discussed, the vanilla scalar quantization using rounding is naturally idempotent. However, the widely adopted mean-shift trick quantization for mean-scale Gaussian entropy model [Minnen et al., 2018] is not guaranteed to be idempotent. As proposed by [Minnen et al., 2020], the mean-shifted quantization can be formulated as

$$Q(\boldsymbol{y}) = \lfloor \boldsymbol{y} - \boldsymbol{\mu} \rceil + \boldsymbol{\mu}, \tag{13}$$

where $\lfloor \cdot \rceil$ is scalar quantization and $\boldsymbol{\mu}$ is the predicted mean of $\boldsymbol{y}$. Let $\boldsymbol{y}_1$ denote the result after the first application of $Q$, then idempotence requires that $Q(\boldsymbol{y}_1) = \boldsymbol{y}_1$, which in turn requires $\boldsymbol{y}_1 - \boldsymbol{\mu}$ to be integer. However, no existing method can meet this requirement, thus $Q$ is not ensured to be idempotent.

We propose two types of circumvention to solve this issue. For the first-type circumvention, we change the quantization into

$$Q_i(\boldsymbol{y}) = \lfloor \boldsymbol{y} - \lfloor \boldsymbol{\mu} \rceil \rceil + \lfloor \boldsymbol{\mu} \rceil \tag{14}$$

By adding $\lfloor \cdot \rceil$ around the predicted mean $\boldsymbol{\mu}$, we force the quantization result to be integer, thus $\boldsymbol{y} - \lfloor \boldsymbol{\mu} \rceil$ is guaranteed to be integer from the second application of $Q_i$, and then $Q_i$ is idempotent.

For the second-type circumvention, we resort to the original definition of mean-scale Gaussian entropy model [Minnen et al., 2018], and calculate the quantized CDF (cumulative distribution function) regarding $Q_i(\boldsymbol{y}) = \lfloor \boldsymbol{y} \rceil$ on the fly during inference.

### 3.4 Overall Framework of Right-Invertible Codec

The overall framework is depicted in Fig. 2. Following prior works [Ballé et al., 2017, 2018, Cheng et al., 2020, He et al., 2022] in LIC, We use the mainstream four-stage framework in our work. Specifically, the encoding transform is divided into four stages, and each stage decreases the resolution by a factor of 2. For the first 3 stages, each stage starts with a right-invertible convolution layer (described in Sec. 3.1) and ends with a c-GDN normalization layer (described in Sec. 3.2). The last stage only consists of one right-invertible convolution layer.

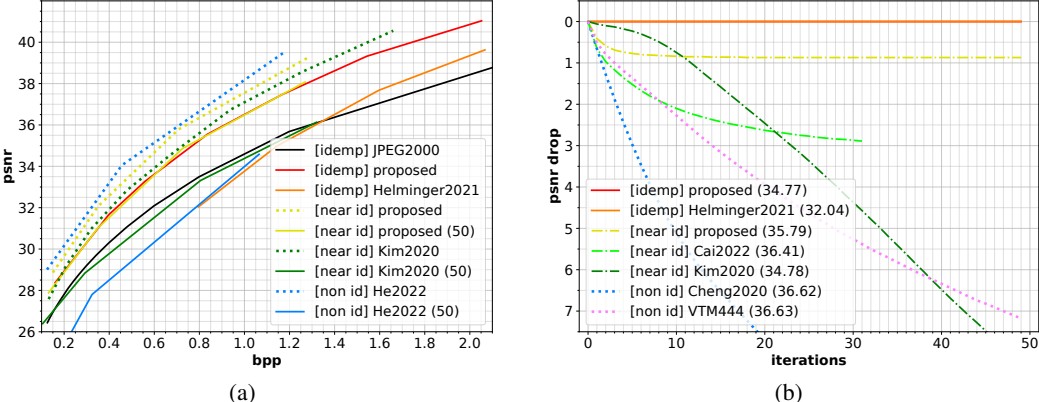

Figure 3: Performance of different codecs on Kodak. Idempotent codecs are marked as `idemp`, near-idempotent codec are marked as `near id` and non-idempotent codecs are marked as `non id`. (a) First-time and re-compression (upto 50 times) RD performance of different codecs. Idempotent codecs only report first-time RD performance (re-compression RD performance is the same). (b) PSNR drop during re-compression (upto 50 re-compression) of different codecs. Note that idempotent codecs are straight lines and cover each other in the figure.

The decoding transform is built to be a right-inverse of the encoding transform. Note that not all layer in the decoding transform has its own weights. Specifically, for c-GDN normalization layers, there is no additional weights since they are the inverse of their counterparts in the encoding transform. For right-invertible convolution layers, the only additional weights appear in the null-space enhancement (Sec. 3.1).

Entropy model does not influence idempotence in the proposed framework, thus we use the off-the-shelf mean-scale Gaussian entropy model [Minnen et al., 2018] for its availability and efficiency. Additionally, we use the right-invertible quantization (described in Sec. 3.3) for quantizing the code.

### 3.5 Extension to Near-Idempotent Learned Image Codec

Idempotent codec can make sure that the RD performance keeps unchanged during any times of re-compression. However, this strict idempotence comes with a price that the decoding transform must be the right-inverse of encoding transform. This limitation reduces the expressiveness of transforms and is empirically harmful to the first-time RD performance.

To adapt to the cases where first-time RD performance also matters, we propose to extend our idempotent codec near-idempotent codec by relaxing the right-invertibility. The relaxation of right-invertibility is simple yet effective: we change the first right-invertible convolution layer (described in Sec.1) to be non-surjective, and keep all the rest layers surjective. This is done by allowing $D$ to be smaller than $d$ for the kernel $K$ in the first right-invertible convolution layer. By keeping the right-invertibility of most layers, our near-idempotent codec is more stable to re-compression than existing near-idempotent codecs [Kim et al., 2020, Cai et al., 2022], while achieving comparable or better first-time RD performance.

## 4 Experiments

### 4.1 Experiment Setup

All the models are trained on the training split of open-images dataset [Kuznetsova et al., 2020], and all the evaluations are conducted on the Kodak dataset [Franzen, 1999].

We sketch the training schedule accordingly from existing literature [Ballé et al., 2017, 2018, Minnen et al., 2018, 2020, Cheng et al., 2020]. Images are randomly cropped to $256 \times 256$ for training, and a batch size of 16 is used. All the models are trained using an Adam optimizer. The learning rate is initially set to $10^{-1}$, and decays by a factor of 10 when plateaued.

We choose four bitrate level accoding to the benchmark setting in [Kim et al., 2020]. Specifically, we set $\lambda = \{18, 67, 250, 932\} \times 10^{-4}$, and models trained with these $\lambda$ reaches average bitrates from 0.2-1.5 on Kodak dataset. Following prior works [Ballé et al., 2017, 2018], we use a smaller code channels (192) for lower-bpp points, and use a bigger code channels (320) for higher-bpp points. The learned function $f(\cdot)$ in Eq.10 is implemented with a residual block.

All the experiments are conducted on a computer with AMD EPYC 7742 64-Core Processor and 8 Nivida A30 GPU. All the code is implemented based Python 3.9, Pytorch 1.12 and CompressAI [Bégaint et al., 2020].

## 4.2 Overall Performance

### 4.2.1 Results of Idempotent Codec

We compare with [Helminger et al., 2021], which is the only prior LIC that achieves idempotent lossy compression to the best of our knowledge. We also compare with traditional idempotent codecs such as JPEG2000.

We report first-time compression RD performance of the above idempotent codecs in Fig. 3(a), and detailed BD-BR and BD-PSNR are listed in Tab. 1. Multi-time re-compression RD performance does not change for idempotent codecs. From the result we see that, our proposed framework exceeds prior art [Helminger et al., 2021] by a large margin, which clearly validates the superiority of right-inverse over strict inverse on the idempotent lossy compression task.

We also compare the FLOPs and encode-decode time in Tab. 1. The results clearly shows that the proposed idempotent framework is also more efficient than [Helminger et al., 2021].

Table 1: The BD-BR, BD-PSNR, FLOPs and encode-decode time of different methods on Kodak dataset. FLOPs and enc-dec time are calcualted on an input of shape $256 \times 256 \times 3$.

| Methods | BD-BR (%) ↓ | BD-PSNR (dB) ↑ | GFLOPs ↓ | time (ms) ↓ |
|---|---|---|---|---|
| *Idempotent Codec* | | | | |
| JPEG2000 | 0.00 | 0.00 | - | - |
| [Helminger et al., 2021] | 4.83 | -0.21 | 15.89 | 185 |
| Proposed Idempotent | -28.75 | 1.63 | 8.40 | 110 |

### 4.2.2 Results of Near-Idempotent Codec

For near-idempotent codecs, we compare against prior work [Kim et al., 2020], as well as [Cai et al., 2022] which utilizes a partially invertible structure. To demonstrate the advantages over non-idempotent codecs, we also compare with non-idempotent learned codecs [Ballé et al., 2018, Cheng et al., 2020, He et al., 2022] as well as traditional codecs BPG and VTM.

Following the benchmark protocol in [Kim et al., 2020], we report the PSNR drop of the above codecs upto 50 re-compression. RD trade-off points whose first-time bpp is closest to but not greater than 0.8 bpp are chosen for each codec. The results is shown in Fig. 3(b) and listed in detail in Tab. 2. From the result we see that, in the near-idempotent setting, the PSNR drop of proposed framework is 0.87dB, whereas [Kim et al., 2020] and [Cai et al., 2022] has more than 2dB PSNR drop. Additionally, the PSNR drop of the proposed framework almost converges within 10 re-compression, while other near-idempotent frameworks still experience evident PSNR drop after 30 or even 50 re-compression. Codecs that do not consider idempotence suffers a much more severe drop of PSNR during re-compression.

We also provide the RD performance of the proposed idempotent and near-idempotent frameworks, as is shown in Fig.3(a). It is clear that near-idempotent framework has much better first-time compression RD performance than idempotent framework. In terms of re-compression RD performance, however, near-idempotent can only reach similar RD performance with much higher computation cost (8.40 GFLOPs v.s. 48.78 GFLOPs in Tab.1 and Tab.2 ).

These results clearly demonstrate that, even if we break the right-invertibility of the first layer in order to get higher first-time RD performance, the performance drop during re-compression is still acceptable and highly controllable, as opposed to prior works [Kim et al., 2020, Cai et al., 2022].

Table 2: PSNR drop during 50 re-compression of different non-idempotent and near idempotent codecs. FLOPs and encode-decode time tested under the same condition as Tab. 1

| | PSNR Drop (dB) ↓ | | | | GFLOPs ↓ | time (ms) ↓ |
|---|---|---|---|---|---|---|
| round = | 5 | 10 | 25 | 50 | | |
| *Non-Idempotent Codec* | | | | | | |
| BPG | 1.16 | 1.93 | 2.10 | 2.19 | - | - |
| VTM | 1.19 | 2.09 | 4.50 | 7.18 | - | - |
| [Ballé et al., 2018] | 2.18 | 3.17 | 5.65 | 8.46 | 6.23 | 80 |
| [Cheng et al., 2020] | 2.44 | 4.76 | 8.59 | 12.40 | 51.99 | >1000 |
| *Near-Idempotent Codec* | | | | | | |
| [Kim et al., 2020] | 0.18 | 0.61 | 3.18 | 8.26 | 6.23 | 80 |
| [Cai et al., 2022] | 1.36 | 2.01 | 2.75 | - | 131.46 | 240 |
| Proposed Near-Idempotent | 0.74 | 0.83 | 0.87 | 0.87 | 48.78 | 115 |

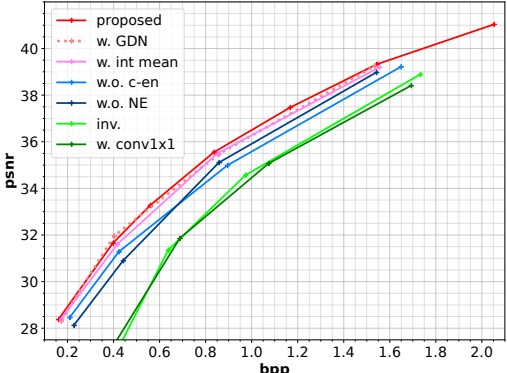

(a) RD performance of different architectures on Kodak

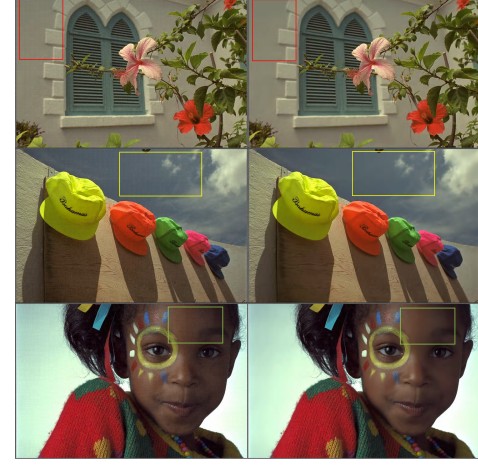

(b) Influence of coupling enhancement (better viewed zoomed-in)

Figure 4: Ablation studies for different architectures. (a) Re-compression RD performance of different idempotent architectures on Kodak dataset (except for w.GDN, for which we report first-time compression as it is non-idempotent). (b) qualitative results for w. (right) or w.o. (left) coupling enhancement.

## 4.3 Ablation Studies

**Inverse v.s. Right-Inverse** The importance of right-inverse for idempotent lossy compression can already be seen from the difference between the proposed framework and [Helminger et al., 2021] in Fig. 3(a) and Tab. 1. To further demonstrate this importance, we test what happens if the proposed framework is changed from right-invertible to invertible. Specifically, we increase the output dimension of the proposed blocked convolution (Sec. 3.1) to its input dimension, so that it becomes fully invertible. We also test what happens if the proposed blocked convolution is replaced with invertible $1 \times 1$ convolution [Kingma and Dhariwal, 2018]. For both cases, the encoding transform is fully invertible.

Fig. 4 (a) shows that both invertible blocked convolution (inv.) and invertible $1 \times 1$ convolution (w.conv1x1) suffer dramatic RD performance degradation compared with the proposed right-invertible framework (proposed). This result further demonstrated that it is important for idempotent LIC to have right-inverse rather than inverse.

**Impact of Null-Space Enhancement** Null-space enhancement is introduced in our framework to enable an adaptive right-inverse for linear layers rather than a fixed Moore-Penrose psudo-inverse. As is shown in Fig. 4, for the two lower-bpp points, the framework with null-space enhancement

(`proposed`) has a 1 dB advantage over the framework without null-space enhancement (`w.o.NE`), whereas for the two higher-bpp points this advantage shrinks to be smaller than 0.5 dB. The reason is that the two lower-bpp points have a smaller code channels (192) than the two higher-bpp points (320). A smaller code channel means a larger null space, and thus null-space enhancement can make larger improvement.

**Impact of Coupling Enhancement** Coupling enhancement improves the receptive field of convolution. The proposed blocked convolution is parallel friendly, but also restricts the receptive field. Qualitatively, this constraint results in obvious deterioration in the flat area, such as the wall, the sky and the face, as is shown in the left column of Fig. 4(b). By introducing coupling enhancement after blocked convolution, the restriction is lifted and the block artifact on the reconstructed images are removed, as is shown in the right column of Fig. 4(b). Quantitatively, the RD performance is also improved by coupling enhancement, which is shown by comparing `proposed` and `w.o.c-en` in Fig. 4(a).

**Impact of Right-Invertible Quantization** As is pointed out in Sec. 3.3, the mean-shifted quantization is not guaranteed to be idempotent, and we propose two alternatives to circumvent this issue. In the proposed framework, we use the second-type circumvention. Here we compare these two choices. As is shown in Fig. 4(a), first-type circumvention (`w.int mean`) performs slightly worse than second-type circumvention (`proposed`). This is because first-type circumvention forces the mean to be integer, whereas second-type does not has this constraint. However, the second-type circumvention requires calculating the quantized CDF on the fly during inference.

**Impact of Right-Invertible GDN** To make the inverse of GDN layer actually computable in the proposed framework, we combine the coupling structure [Papamakarios et al., 2021] with GDN, as is described in Sec. 3.2. Here we demonstrate that such workaround does not affect the RD performance. Specifically, we change the c-GDN and c-GDN$^{-1}$ layer in our framework (Fig. 2(b)) back to GDN and iGDN layer [Ballé et al., 2015]. Note that this change makes the framework non-idempotent, and is only used to test the RD performance. From the results in Fig. 4(a) we see that, Whether to use coupling GDN (`proposed`) or original GDN (`w.GDN`) has negligible influence on RD performance. Thus coupling GDN is an efficient and effective replacement for GDN in idempotent compression framework.

# 5   Related Work

The idempotence has been a crucial consideration for lossy image codecs. For image codecs without prediction encoding like JPEG [Wallace, 1991] and JPEG2000 [Taubman et al., 2002], idempotence is naturally assured as invertible encoding transforms like DCT or DWT is adopted. As long as the quantization is idempotent (which is true for scalar quantization), the whole codec becomes idempotent [Joshi et al., 2000, Richter et al., 2017].

The idempotence of learned image compression is firstly studied by [Kim et al., 2020], which proposes a near-idempotence solution that alleviates re-compression loss but does not eliminate it. [Cai et al., 2022] further improves over [Kim et al., 2020] while it is not able to achieve strict idempotence. [Helminger et al., 2021] is the first LIC that achieves idempotence, using fully invertible normalizing flow as encoding transform. However, it's RD performance dramatically falls behind modern LIC.

Our work is also related to SurVAE flow [Nielsen et al., 2020]. On the one hand, the idempotence of LIC requires the encoding transform to be surjective, which is similar to the goal of SurVAE flow. The difference is, we consider deterministic right inverse, whereas SurVAE flow considers stochastic right inverse. On the other hand, the techniques proposed in this paper, such as blocked convolution and null-space enhancement, can be used to improve SurVAE flow.

# 6   Discussion & Conclusion

To conclude, we first identify that invertibility is sufficient but not necessary for idempotent codec, and it can be instead relaxed to right-invertibility. Based on this identification, we investigate the practical implementation of right-inverse with efficiency and expressiveness. Empirically, We show that the proposed method achieves state-of-the-art RD performance among idempotent codecs. Furthermore,

our codec can be easily relaxed into a near-idempotent codec, which also achieves state-of-the-art re-compression performance among near-idempotent codecs.

For future work, one possible direction is constructing right-invertible transform without function composition. Currently, the right-invertible transform is constructed using composition of surjections. Such construction strictly restricts the latent dimension to be non-increasing throughout the transform. This restriction is conflict with the prevalent design logic of neural network and is detrimental to expressiveness. It would be interesting to see if this restriction could be removed and how much the performance could be improved.

## Acknowledgements

Funded by Baidu Inc. through Apollo-AIR Joint Research Center.

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

# Appendices

# A Additional Explanation for the Methods

## A.1 Complexity Analysis of Block Convolution

To discuss this problem, consider a 2-dimension convolution with kernel size $K \times K$, stride $S \times S$, input spatial size $H \times W$, input channel $C_i$ and output channel $C_o$. Assuming the in-place parallel matrix multiplication[Randall, 1998] is used.

For serial method, as is described in Line 104-109, we first subtract the influence of the already-solved pixels, which takes $O(C_i K^2)$ time. Then we solve for the rest pixels, which takes $O(C_o)$ time. The point is, this subtract-then-solve procedure needs to be done $O(HW/S^2)$ times, in serial, so the overall complexity is $O(\frac{HW}{S^2}(C_i K^2 + C_o))$.

For parallel method implemented with the proposed blocked convolution, we can solve for all pixels with $O(C_o)$ time complexity.

We can see that, compared with the parallel method, time complexity of serial method is forbiddingly higher and not viable for practical usages. In terms of re-compression performance, both serial method and parallel method are right-inverse, so idempotence can be achieved for both.

## A.2 More on Null-space Enhancement

**Parameterization of Surjective Linear Transform**

For a linear transform with kernel $K \in \mathbb{R}^{D \times d}$ to be surjective, $K$ must have full column rank, i.e., $d$ must be less or equal to $D$ and the rank of $K$ is $d$. The vanilla parameterization of $K$ as a matrix cannot ensure this property. Thus, we use singular value decomposition (SVD) parameterization to ensure that $K$ is surjective. Specifically, $K$ is decomposed as $K = USV^T$, where $U$ is a $D \times d$ orthonormal matrix, $S$ is a $d \times d$ diagonal matrix with non-zero diagonal elements, and $V$ is a $d \times d$ orthonormal matrix. With this decomposition, $K$ is guaranteed to have full column rank, and is thus surjective. Accordingly, Eq. 10 is parameterized as

$$X = YVS^{-1}U^T + f(Y)(I - UU^T) \tag{15}$$

where $S^{-1}$ is the inverse of non-zero diagonal matrix $S$. For orthonormal matrix $U$ and $V$, we adopt the parameterization in [Lezcano-Casado, 2019]. The matrix $S$ is diagonal and trivial to parameterize. To avoid arithmetic overflow, we restrict the diagonal elements of $S$ to be within $[0.1, 10]$.

**Derivation of Null-space equality**

$\forall F \in \mathbb{R}^{b \times D}$, $X = YK^+K + F(I - KK^+)$ is a solution to $Y = XK$.

*Proof.* From the property of Moore-Penrose pseudo-inverse [Moo] we know that

$$KK^+K = K. \tag{16}$$

Additionally, for a orthonormal $K$ (with linearly independent columns), we have

$$K^+K = I. \tag{17}$$

Then, right-multiply $X(Y; K)$ by $K$, we get

$$XK = YK^+K + F(I - KK^+)K \tag{18}$$
$$= Y + FK - FK = Y.$$

$\square$

# B More Experimental Results

## B.1 More Experiment Setup

The detailed encoding and decoding transform is illustrated in Fig. 5. To extend the idempotent framework to near-idempotent framework, we change the first blocked convolution in the encoding

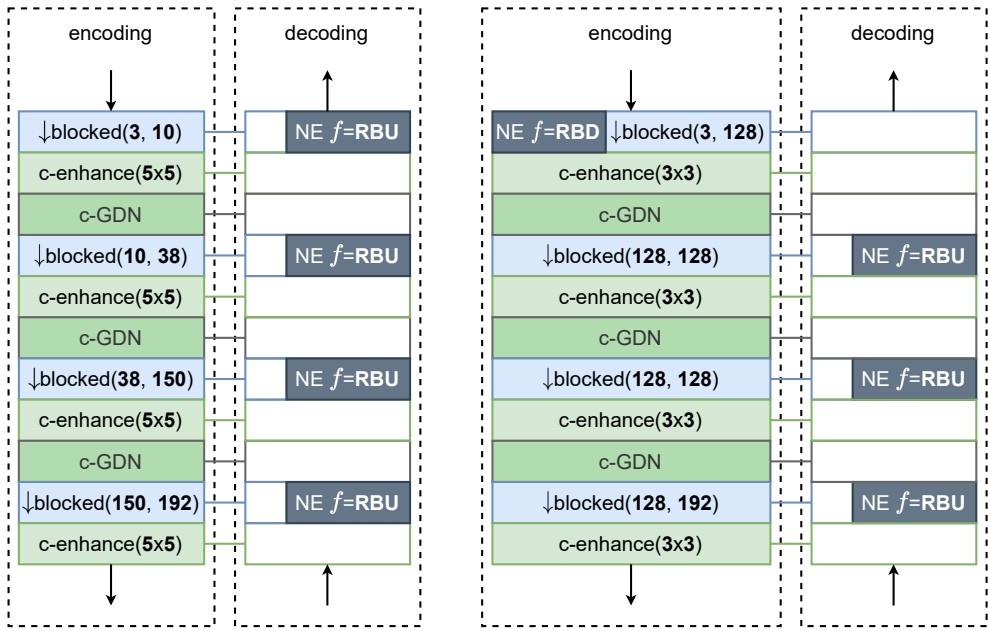

(a) Details of the proposed imdempotent framework   (b) Details of the proposed near-imdempotent framework

Figure 5: Detailed encoding and decoding transform for the proposed idempotent and near-idempotent framework. **blocked** refers to the proposed blocked convolution. The (input, output) channels are annotated in the brackets, and stride is annotated using down-arrow. **NE** refers to the proposed null-space enhancement, and $f$ is learned parametric function. Here RBU/RBD are the residual block used for upsampling/downsampling (Fig. 6(b)), respectively. **c-enhance** refers to the proposed coupling enhancement using coupling structure (Fig. 6(a)). The kernel size of the convolution is annotated in the brackets. **c-GDN** refers to the proposed right-invertible normalization using coupling structure (Fig. 6(a)). Blanked rectangular refer to the right-inverse/inverse of the corresponding layer, and has no additional weights.

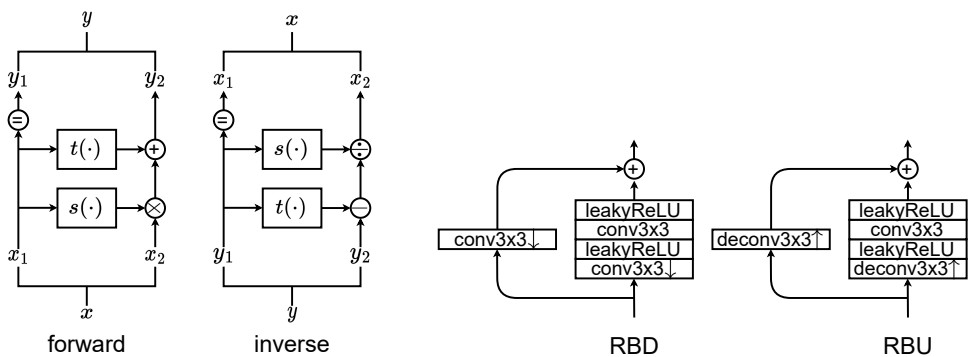

(a) Couping structure used in c-enhence and c-GDN   (b) Residual block used in null-space enhancement

Figure 6: Submodules used in the proposed framework: (a) coupling structure used in c-enhance and c-GDN; (b) residual block downsampling (RBD) and residual block upsampling (RBU) used in the $f(\cdot)$ of null-space enhancement.

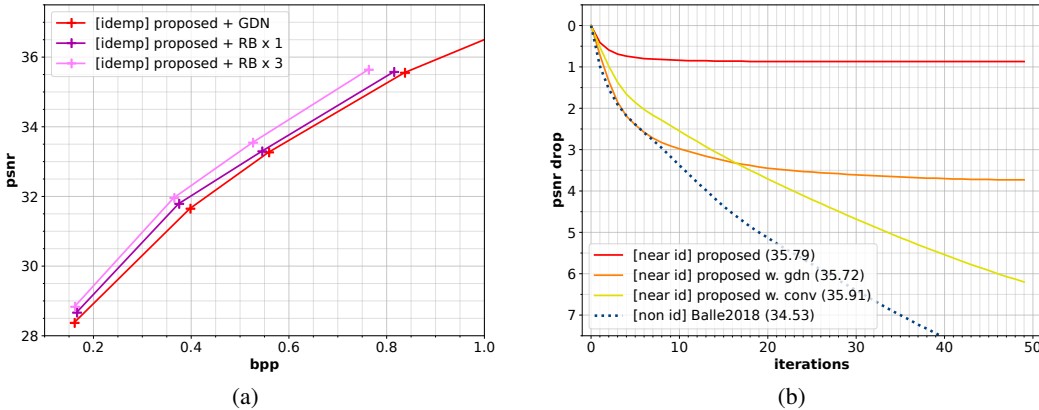

(a)                  (b)

Figure 7: Additional experiments on: (a) extendability of the proposed idempotent framework (b) functionality of different components of the proposed near-idempotent framework.

transform to non-surjective by increasing its output channel number from 10 to 128. Since this blocked convolution is no longer surjective, it is no longer right-invertible. However, its corresponding layer in the decoding transform can be make surjective and right-invertible. Thus, we make its corresponding layer in the decoding transform surjective, and use the null-space enhancement on the encoding side.

Coupling structure [Dinh et al., 2016] used in coupling enhancement (c-enhance) and coupling GDN (c-GDN) is illustrated in Fig. 6(a). For c-enhance, the scale $s(\cdot)$ and translation $t(\cdot)$ are convolution. For c-GDN, the scale $s(\cdot)$ and translation $t(\cdot)$ are GDN [Ballé et al., 2015]. Following the usage guideline in [Dinh et al., 2016], we concatenate two coupling structure with the opposite way of splitting in one c-enhance/c-GDN.

We provide additional experiments on (a) extendability of the proposed idempotent framework (b) functionality of different components of the proposed near-idempotent framework. Specifically, to demonstrate the extendability of the proposed idempotent framework, we replace GDN with residual blocks as suggested by [He et al., 2022] (RB×1 for one residual block, or RB×3 for three residual blocks), and report the RD performance in Fig.7(a). It is clear that, replacing GDN with the more recent residual blocks would also improve our framework to a similar degree. Thus our proposed framework is compatible with the recent advances in LIC and has good extendability.

To analysis the functionality of different components of the proposed near-idempotent framework, we test what happens if the modification to each component is not applied, and report the PSNR drop during re-compression in Fig.7(b). Specifically, we test keeping the GDN layers unchanged (w. gdn) or keeping the convolution layers unchanged (w. conv). Keeping both the GDN layers and the convolution layers would reduce to baseline Balle2018. The results shows that keeping more layers unchanged may slightly improve first-time compression performance, but is evidently harmful to re-compression performance. The proposed near-idempotent framework has the best re-compression performance among these settings.

We include the implement of other codecs as follows. For codecs that have open-source implementations, we use that implementation. For codecs that do not have open-source implementations, we either use the data provided in the paper, or re-implement by ourselves if the detailed architecture is provided.

- Implementations from CompressAI [Bégaint et al., 2020]: Balle2017[Ballé et al., 2017], Balle2018[Ballé et al., 2018], Cheng2020[Cheng et al., 2020], JPEG2000[Taubman et al., 2002], BPG444[Sullivan et al., 2012], VTM444[Bross et al., 2021]

- Data from the original papers: Helminger2021[Helminger et al., 2021], Cai2022[Cai et al., 2022]

- Our re-implementation: Kim2020[Kim et al., 2020]. Specifically, we re-implement the FI loss proposed in this work on Balle2018 [Ballé et al., 2018].

### B.2 More Quantitative & Qualitative Results

See Fig. 8-11 for more quantitative results.
See Fig. 12-15 for more qualitative results.

## C More Discussion

### C.1 Limitation

In this work, the surjective encoding transform is constructed using function composition of simple surjections. This construction strategy limits the latent dimension to be non-increasing throughout the encoding transform. This limitation contradicts the mainstream design logic of neural network, and is harmful to expressiveness.

A function composition of surjections is always a surjection, but a surjection needs not to be a function composition of surjections [Mac Lane, 2013]. Thus this restriction could be lifted by more advanced construction strategy of surjection.

### C.2 Broader Impact

Improve the rate-distortion of re-compression has positive social impact. Re-compression constantly happens in the transmission and redistribution of image data. Reducing the bitrate can save the resources, energy and the carbon emission during these processes.

### C.3 Reproducibility Statement

All theoretical results are proven in Appendix. A. For experimental results, all the datasets used are publicly available, and the implementation details are provided in Appendix. B. Furthermore, the source code for reproducing experimental results are provided in supplementary materials.

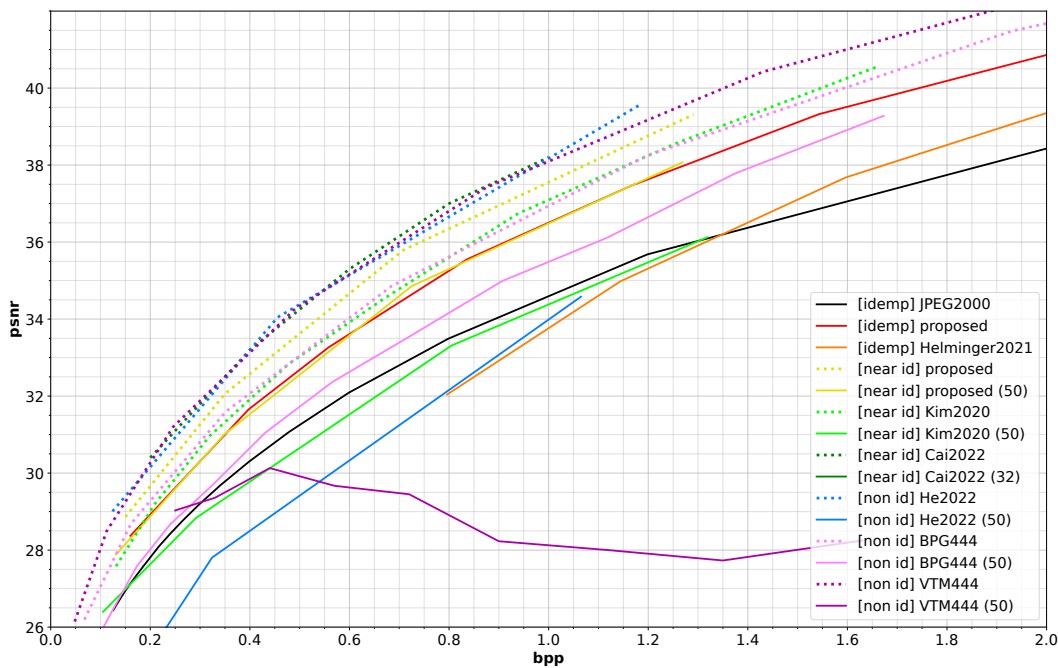

Figure 8: **PSNR-BPP** curve on **Kodak**. Idempotent codecs are marked as `idemp`, near-idempotent codec are marked as `near id` and non-idempotent codecs are marked as `non id`. First-time compression performance is plotted in dotted line, and re-compression performance (upto 50 times) is plotted in solid line.

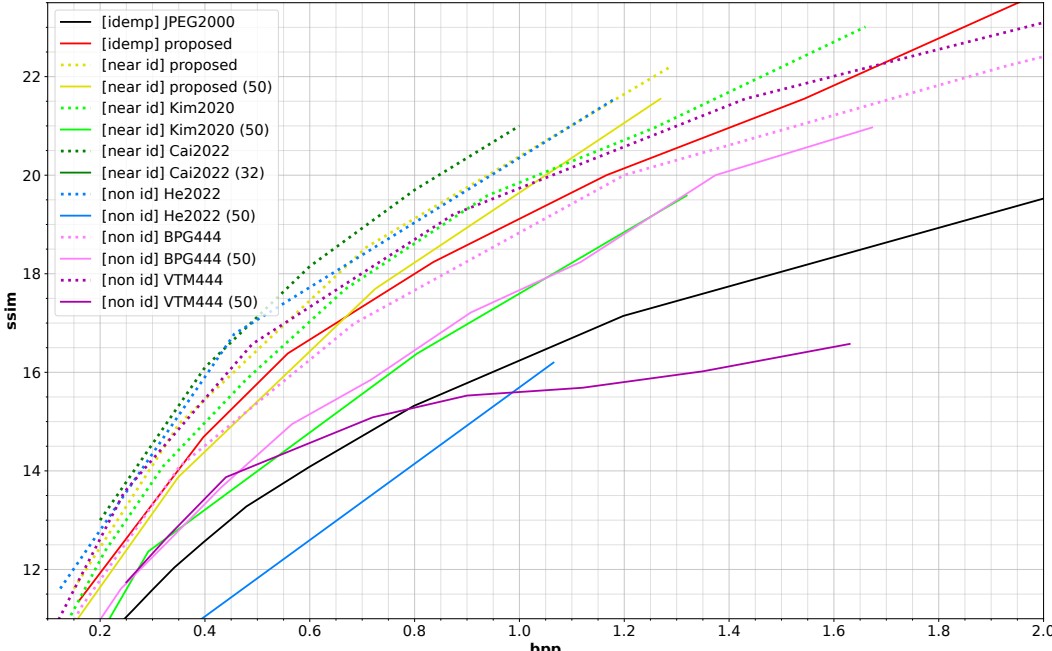

Figure 9: **MSSSIM-BPP** curve on **Kodak**. Idempotent codecs are marked as `idemp`, near-idempotent codec are marked as `near id` and non-idempotent codecs are marked as `non id`. First-time compression performance is plotted in dotted line, and re-compression performance (upto 50 times) is plotted in solid line. All models are optimized for minimizing MSE.

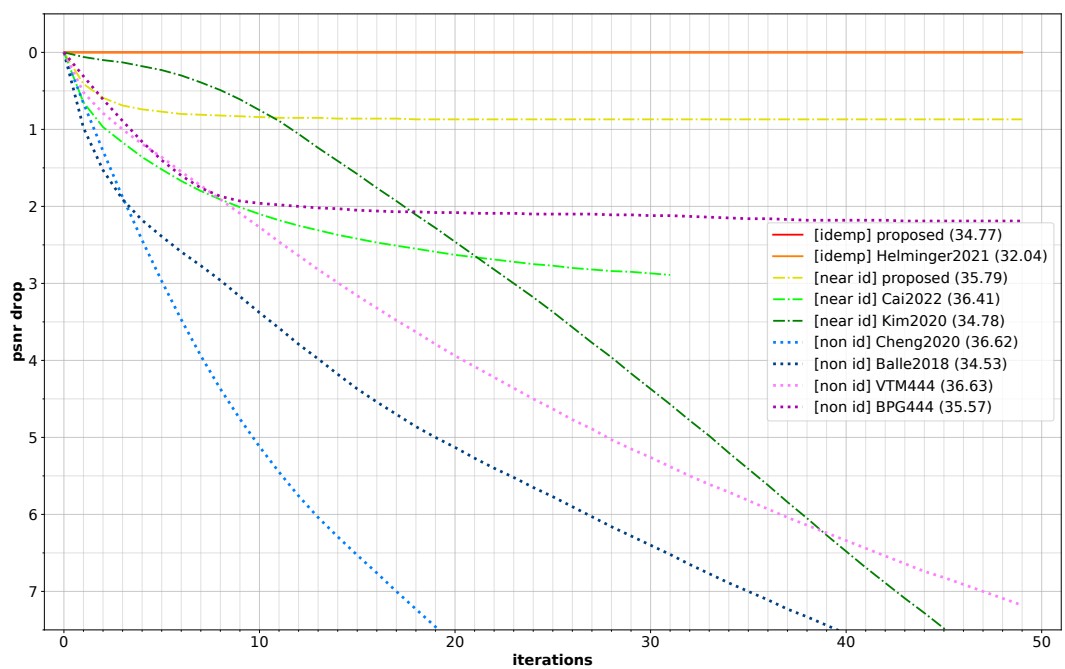

Figure 10: **PSNR** drop upto 50 re-compression on **Kodak**. Idempotent codecs are marked as `idemp`, near-idempotent codec are marked as `near id` and non-idempotent codecs are marked as `non id`. First-time PSNR is annotated in (·).

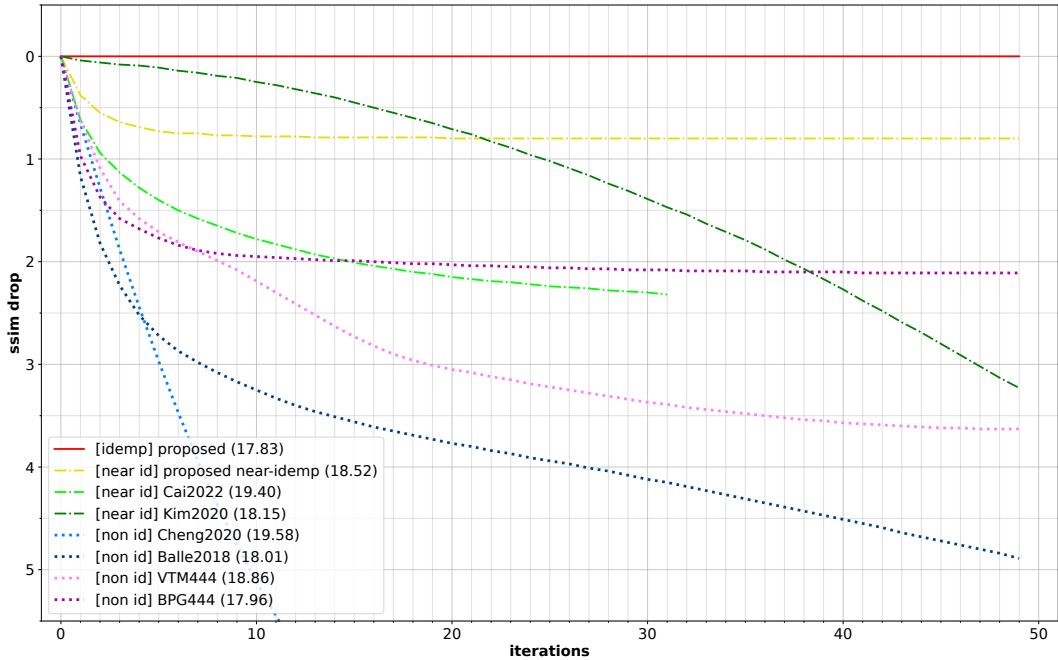

Figure 11: **MS-SSIM** drop upto 50 re-compression on **Kodak**. Idempotent codecs are marked as `idemp`, near-idempotent codec are marked as `near id` and non-idempotent codecs are marked as `non id`. First-time MS-SSIM is annotated in (·). All models are optimized for minimizing MSE.

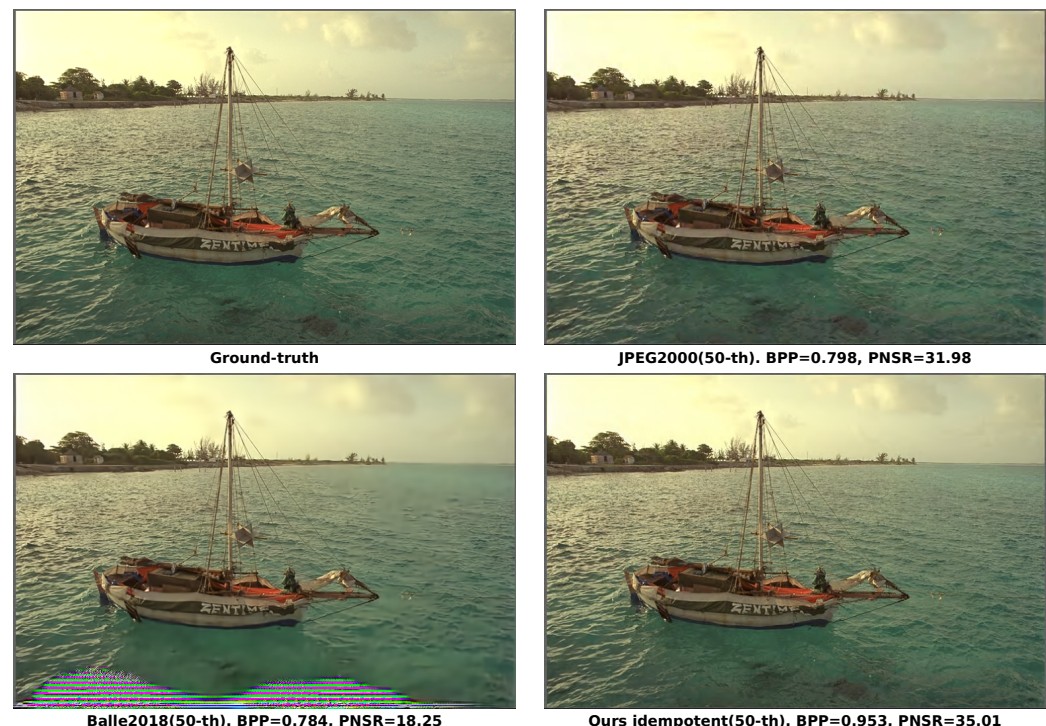

Figure 12: Qualitative comparison on reconstructed kodim06 image after 50 times re-compression.

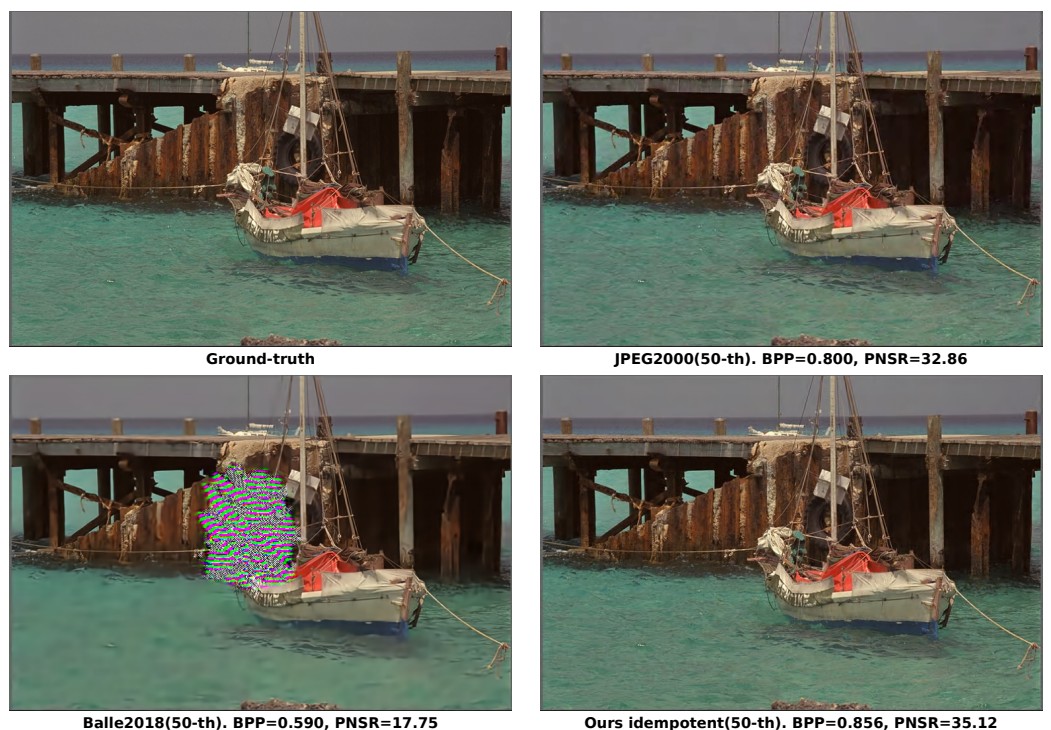

Figure 13: Qualitative comparison on reconstructed kodim11 image after 50 times re-compression.

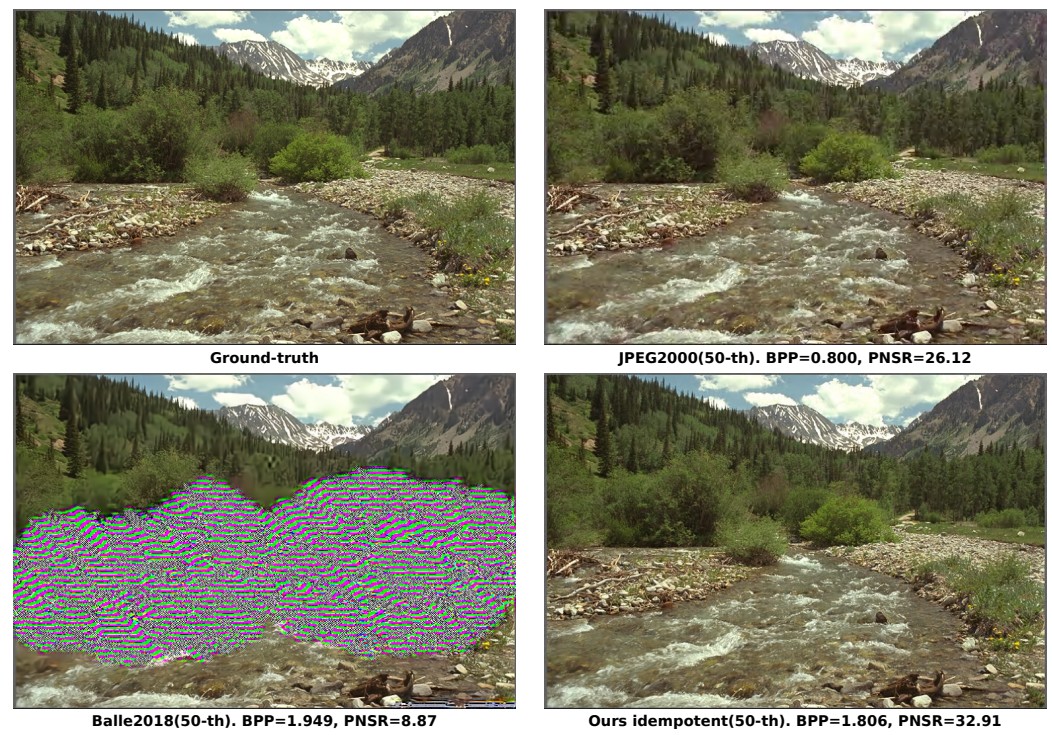

Figure 14: Qualitative comparison on reconstructed `kodim13` image after 50 times re-compression.

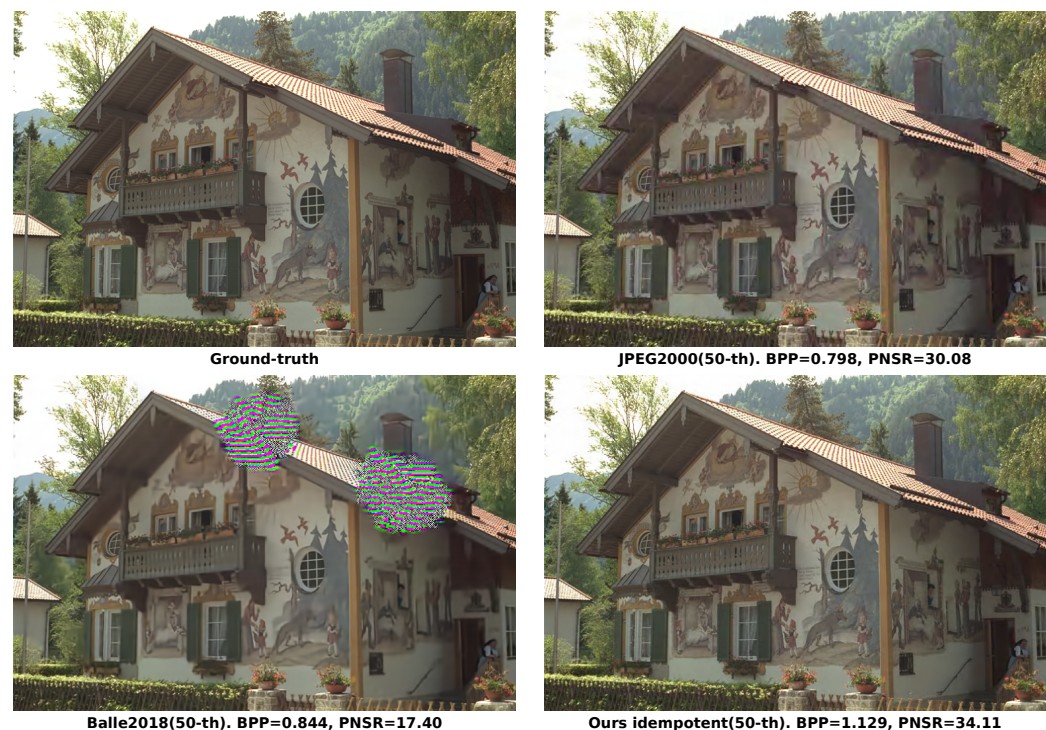

Figure 15: Qualitative comparison on reconstructed `kodim24` image after 50 times re-compression.

