# OpenReview forum: "Idempotent Learned Image Compression with Right-Inverse"
_NeurIPS.cc/2023/Conference — NeurIPS 2023 poster_

### Official Review · Reviewer_fD4t · 2023-06-22

**Soundness:** 3 good
**Presentation:** 2 fair
**Contribution:** 2 fair
**Rating:** 6
**Confidence:** 3

**Summary:**

The paper shows how to achieve a truly idempotent image compression method, where f(x) = f(f(x)). The paper shows that it's sufficient to have E(D(y)) == y. This only requires a surjective E.

**Strengths:**

Interesting derivation of the required conditions for idempotence. Also nice to see this problem studied.

**Weaknesses:**

- Old baselines: I think GDN is barely used in SOTA image codecs, where it's all ReLU or Leaky ReLU. It would have been interesting to see this method applied to eg. ELIC from 2022 vs. Balle's 2018 method.
- The linear algebra presentation was hard to follow (p4). Some more intuitive insights like in the introduction would have been useful.


Minor: Typo in section 3 header ("Invertibie")

**Questions:**

-

**Limitations:**

Only applied to old architectures from 2018 which contain 4 conv layers and GDN.

---

> ### Author Rebuttal · Authors · 2023-08-09
>
> # To reviewer fD4t
>
> Thanks for your advices. We address your concerns as follows.
>
> **weakness-1 & limitation**: We test replacing GDN with residual blocks (as suggested by ELIC[1]), and report as follows (also shown in **[rebuttal fig.4]**):
>
> | framework | BD-rate (RB $\times$ 1 v.s. GDN) | BD-rate (RB $\times$ 3 v.s. GDN) |
> | -------- | -------- | -------- |
> | [idemp] proposed  | -2.61     | -9.28     |
> | Balle2018         | -2.35     | -10.32    |
> | Minnen2018        | -1.36     | -8.97     |
>
> The results of Balle2018 and Minnen2018 are from Tab.4 in  [1], e.g. for Balle2018 RB  $\times$ 1 v.s. GDN the BD-rate is $[(100 + 5.68) / (100 + 8.23) - 1] \times 100  = -2.35$.
>
> It is clear that, replacing GDN with the more recent residual blocks would also improve our framework to the same degree. Thus our propsed framework is compatible with the recent advances in LIC.
>
> **weakness-2**: Thanks for your kind advice. We will make these parts more clearly presented.
> Specifically, we will first describe what does right-invertible linear operations look like, then we explain why null-space decomposition preserses the right-invertibility, and finally we introduce how we conduct null-space enhancement.
>
> [1] He, Dailan, et al. "Elic: Efficient learned image compression with unevenly grouped space-channel contextual adaptive coding." CVPR, 2022.

---

> > ### Comment · Reviewer_fD4t · 2023-08-14
> >
> > Thank you for the rebuttal. I will stick with my accept rating.

---

### Official Review · Reviewer_4J8x · 2023-07-05

**Soundness:** 3 good
**Presentation:** 3 good
**Contribution:** 3 good
**Rating:** 5
**Confidence:** 3

**Summary:**

The paper argues that invertibility is sufficient but not necessary for achieving idempotent codecs, and proposes a framework for achieving idempotent learned image compression (LIC) with right-inverse, which allows more flexible and expressive transforms. This paper details the expressive and right-reversible atomic transformations used in LIC, including convolution (using blocked rearrangement and null-space enhancement), normalization, and quantization.

**Strengths:**

1、The paper first theoretically proves that idempotence can be relaxed to be right reversible and details the Right-Invertibie Codec.
2、The proposed framework achieves state-of-the-art RD performance among idempotent codecs. Also, it can be easily relaxed into a near-idempotent codec, which also achieves state-of-the-art re-compression performance.
3、The paper is well organized with detailed description of various parts of the whole right-reversible atomic transformations.


**Weaknesses:**

1、Lack of analysis of the computational complexity of the proposed framework: The paper does not provide a detailed analysis of the computational complexity of the proposed framework. While the paper mentions that the proposed framework is efficient and parallel-friendly, it does not provide a detailed analysis of the computational cost of the various components of the framework.
2、The First-time compression RD performance of the proposed codec dramatically falls behind modern LIC. For most scenarios, the first compression is also important. It will be interesting for the author to provide an analysis on feasibility to further improve RD performance of the proposed framework.
3、Limited discussion of the limitations of the proposed framework: While the paper briefly discusses the limitations of the proposed framework.  For example, the paper could discuss the potential impact of the assumption that the input image is preprocessed to have zero mean and unit variance on the performance of the framework.
4、Lack of comparison with non-idempotent codecs: The paper only compares the proposed framework with other idempotent and near-idempotent codecs.
5、Limited analysis of the impact of null-space enhancement: The paper only shows the impact of null-space enhancement on two lower-bpp points and two higher-bpp points, and does not provide a more comprehensive analysis of the impact of this technique on the performance of the framework.


**Questions:**

1、Compared with the serial method, how much time complexity does the blocked rearrangement save, and will it have a huge impact on RD performance and re-compression performance?
2、Can the convolution, GDN, quantization, and other modules designed in the paper with the Right-Invertibie properties be applied to existing learned image compression frameworks? In that case, will the performance of the initial compression be dropped excessively?
3、The subjective encoding transform limits the expressiveness of the network, try other better mapping strategies?
4、Could the paper discuss the potential impact of the assumption that the input image is preprocessed to have zero mean and unit variance on the performance of the framework?


**Limitations:**

1. The proposed framework considers compression on integer latent, which is different from the common LIC method.
2. The assumption that the input image is preprocessed to have zero mean and unit variance may not hold for all images.

---

> ### Author Rebuttal · Authors · 2023-08-09
>
> # To reviewer 4J8x
>
> Thanks for your advices. We address your concerns as follows.
>
> **weakness-1**: We provide the flops of various components of the proposed idempotent framework (for a $256 \times 256 \times 3$ input) as follows.
>
> | components | GFLOPs |
> | -------- | -------- |
> | blocked convolution     | 0.98     |
> | null-space enhancement  | 3.12     |
> | coupling enhancement    | 3.19     |
> | right-invertible GDN    | 0.24     |
> | hyperprior and others   | 0.87     |
> | *overall*               | 8.40     |
>
>
> **weakness-2**: To further improve the RD performance, we can first investigate the surjective mapping restriction. We currently concat multiple surjective mappings to form a big and complex encoding transform, in order to make this encoding transform right-invertible as a whole. However, this 'concat of surjective mappings' is sufficient but not necessary, and damages the expressiveness. It is possible to ensure the right-invertibility of encoding transform with other options, for example using multi-branch structures.
>
> Additionally, the building blocks such as GDN can be improved, for example changed to residual blocks. We use GDN here only to conduct a fair comparison against baseline Balle2018. We test replacing GDN with the more recent residual blocks and see similar improvement as reported in prior works, as is shown in **[rebuttal fig.4]**.
>
> **weakness-3 & question-4 & limitation-2**: We do not make this assumption.
>
> **weakness-4**: in Fig.3, we compare with the non-idempotent traditional codecs {JPEG2000, BPG444, VTM444} as well as learned codecs {Balle2017, Balle2018, Cheng2020}. We will make this more clear.
>
> **weakness-5**: We provide the BD-rate and BD-PSNR of various components of the framework, including null-space enhancement, as follows.
>
>
> | setting | BD-rate ($\downarrow$) | BD-PSNR ($\uparrow$) |
> | -------- | -------- | -------- |
> | proposed    |  0   %     |  0        |
> | w. GDN      |  0.24%     | -0.01     |
> | w. int mean |  4.11%     | -0.20     |
> | w.o. c-en   | 16.48%     | -0.75     |
> | w.o. NE     | 19.68%     | -0.91     |
> | inv.        | 56.03%     | -2.31     |
> | w. conv1x1  | 60.65%     | -2.51     |
>
>
> It is clear that these components are crucial to the proposed framework.
>
> **question-1**: To discuss this problem, consider a 2-dimension convolution with kernel size $K \times K$, stride $S \times S$, input spatial size $H \times W$, input channel $C_i$ and output channel $C_o$. Assuming the in-place parallel matrix multiplication[1] is used. For serial method, as is described in Line 104-109, we first substract the influence of the already-solved pixels, which takes $O(C_iK^2)$ time. Then we solve for the rest pixels, which takes $O(C_o)$ time. This substract-then-solve procedure needs to be done $HW/S^2$ times, so the overall complexity is $O(\frac{HW}{S^2}(C_iK^2+C_o))$. For parallel method implemented with the proposed blocked rearrangement, we can solve for all pixels with $O(C_o)$ time complexity. We can see that, compared with the parallel method, time complexity of serial method is forbiddenly higher and not viable for practical usages.
> In terms of re-compression performance, both serial method and parallel method are right-inverse, so idempotence can be achieved for both.
> In terms of RD performance, the serial method could be better because it does not suffer from block artifact. However, this issue has been addressed by the proposed coupling enhancement.
>
> **question-2**: The proposed idempotent frameworks is implemented based on the existing learned image compression framework Balle18[2], and the performance of the initial compression is not dropped excessively, as is shown in Fig.3 (a).
>
> **question-3**: To our best knowledge, surjective encoding transform is the only way to make encoding transform right-invertible, and this right-invertibility is the key to make the coding process idempotent.
>
> Relaxation of this surjection constraint while keeping right-invertibility would make a promising imporvement over this work.
>
> **limitation-1**: The latents can be made non-integer. For example, we can add {0.1, 0.2, ..., 0.9} as quantization level. In most LIC works the quantized latent is integer (e.g. Balle18[2]).
>
> [1] Randall, Keith H. (1998). Cilk: Efficient Multithreaded Computing (PDF) (Ph.D.). Massachusetts Institute of Technology. pp. 54–57
> [2] Ballé, Johannes, et al. "Variational image compression with a scale hyperprior." International Conference on Learning Representations. 2018.

---

> > ### Comment · Reviewer_4J8x · 2023-08-18
> > **More comments**
> >
> > Some concerns have been addressed by the authors rebuttal. Is it possible to provide some analysis on time complexity of the algorithm? And is it easy to apply the designed modules in this paper to other SOTA learning based codec in a plug-and-play way?

---

> > > ### Author Response · Authors · 2023-08-19
> > >
> > > > Is it possible to provide some analysis on time complexity of the algorithm?
> > >
> > > We extend Tab.1 and Tab.2 in the paper as follows, adding the actual 'encode-decode' time column as reference.
> > >
> > > **Table 1**: The BD-BR, BD-PSNR, FLOPs and encode-decode time of different methods on Kodak dataset. FLOPs and enc-dec time are calcualted on an input of shape 256 × 256 × 3.
> > >
> > > | Method | BD-BR($\%$)$\downarrow$ | BD-PSNR(dB)$\uparrow$ | GFLOPs$\downarrow$ | enc-dec time(ms) $\downarrow$ |
> > > | -------- | -------- | -------- | -------- | -------- |
> > > | JPEG2000 | 0.00 | 0.00 | - | - |
> > > | Helminger2021[1] | 4.83 | -0.21 | 15.89 | 185 |
> > > | Idempotent Proposed | **-28.75** | **1.63** | **8.40** | **110** |
> > >
> > > **Table 2**: PSNR drop during 50 re-compression of different non-idempotent and near idempotent
> > > codecs. FLOPs and encode-decode time tested under the same condition as Tab. 1
> > >
> > >
> > > | Method | PSNR drop (dB) $\downarrow$, round=5 | round=10 | round=25 | round=50| GFLOPs $\downarrow$ | enc-dec time(ms) $\downarrow$ |
> > > | -------- | -------- | -------- | -------- | -------- | -------- | -------- |
> > > | (Non-Id) BPG | 1.16 | 1.93 | 2.10 | 2.19 | - | - |
> > > | (Non-Id) VTM | 1.19 | 2.09 | 4.50 | 7.18 | - | - |
> > > | (Non-Id) Balle18[2] | 2.18 | 3.17 | 5.65 | 8.46 | 6.23 | 80 |
> > > | (Non-Id) Cheng20[3] | 2.44 | 4.76 | 8.59 | 12.40 | 51.99 | >1000 |
> > > | -------- | -------- | -------- | -------- | -------- | -------- | -------- |
> > > | (Near-Id) Kim20[4] | **0.18** | **0.61** | 3.18 | 8.26 | **6.23** | **80** |
> > > | (Near-Id) Cai22[5] | 1.36 | 2.01 | 2.75 | - | 131.46 | 240 |
> > > | Near-Idempotent Proposed | *0.74*  | *0.83* | **0.87** | **0.87** | *48.78* | *115* |
> > >
> > > It is clear that for both the idempotent and near-idempotent setting, the proposed framework achieves SOTA performance with comparable FLOPs and time complexity.
> > >
> > > >  Is it easy to apply the designed modules in this paper to other SOTA learning based codec in a plug-and-play way?
> > >
> > > Yes, these modules can be applied in a plug-and-play way to many other SOTA architectures like [6].
> > >
> > > - For blocked rearrangeed convolution, the only key point is that the receptive field is non-overlapping. Everything else is the same as normal convolution at inference time.
> > > - Null-space enhancement can be applied to trained linear tranforms (including convolution) without changing their weights.
> > > - Coupling enhancement is already plug-and-play.
> > > - Right-invertible activations can be improved from GDN-based to the SOTA residual-block-based [6], as is shown in **[rebuttal fig.4]**.
> > > - Right-invertible quantization can be used in replace for normal quantization in a plug-and-play way.
> > >
> > > However, it is not clear how to apply similar idea to transformer based architecture like [7] in an efficient way. Thanks for raising this point, we think this leaves a meaningful future research direction on how to make transformer based architecture idempotent. We will add those discussions in the final version to inspire more research.
> > >
> > > [1] L. Helminger, et al. "Lossy image compression with normalizing
> > > 353 flows." ICLRW 2021.
> > > [2] J. Ballé, et al. "Variational image compression with a scale hyperprior." ICLR, 2018.
> > > [3] Z. Cheng, et al. "Learned image compression with discretized gaussian mixture likelihoods and attention modules." CVPR, 2020.
> > > [4] J.-H. Kim, et al. "Instability of successive deep image compression." ACM MM, 2020
> > > [5] S. Cai, et al. "High-fidelity variable-rate image compression via invertible activation transformation." ACM MM, 2022
> > > [6] D. He, et al. "Elic: Efficient learned image compression with unevenly grouped space-channel contextual adaptive coding." CVPR, 2022.
> > > [7] Zhu, Yinhao, Yang Yang, and Taco Cohen. "Transformer-based transform coding." ICLR. 2021.

---

### Official Review · Reviewer_H19p · 2023-07-08

**Soundness:** 2 fair
**Presentation:** 2 fair
**Contribution:** 1 poor
**Rating:** 5
**Confidence:** 4

**Summary:**

This paper introduces a learned image codec with a right-inverse transform. The task is to ensure that an image can be re-compressed multiple times without significant quality degradation, while the compression is still lossy. This work is one of the few early attempts along this line of research. Its applications are rather niche.

**Strengths:**

The paper is easy to follow and the task is novel.

**Weaknesses:**

(1) The main idea of null space decomposition has been proposed in J. Schwab, S. Antholzer, and M. Haltmeier,  Deep null space learning for inverse problems: convergence analysis and rates. Inverse Problems, 35(2):025008, 2019.

(2) The necessity and benefits of the blocked rearrangement convolution with coupling enhancement over the ordinary convolution are not clear. It seems to me that this part is not essential to null space decomposition. The authors argue that blocked rearrangement convolution can work more efficiently. However, there is no evidence or ablation study to support the argument.

(3) The right inverse requires E o D to be an identity matrix. However, Eqs. (8) and (9) suggests D o E is an identify matrix.

(4) Can the re-compression be done on different computation platforms while maintaining the right inverse property?

(5) There is no ablation study on the near-idempotent idea. In Section 3.5, the authors mention that the near-idempotent codec has better first-time compression performance than the idempotent codec. Is there any insight into this? Also, how about the recompression performance as compared to the idempotent design?

(6) The proposed method involves SVD. I wonder if this complicates the training process.

(7) The authors should cite "https://github.com/mahaichuan/Versatile-Image-Compression" for invertible compression backbone design.

**Questions:**

See my comments in the weakness section.

**Limitations:**

Please clarify whether the training would be complicated by the use of SVD.

---

> ### Author Rebuttal · Authors · 2023-08-09
>
> # To reviewer H19p
>
> Thanks for your advices. We address your concerns as follows.
>
> **weakness-1**: Yes, and we have mentioned it at Line 43. We are the first to apply null-space decomposition to LIC task. More importantly, we propose several novel designs to enable efficient idempotent LIC as summarized in L39-47.
>
> **weakness-2**: Null-space decompostion is of use only when the right-inverse can be practically calculated. Ordinary convolutions have overlapping receptive fields, and this makes the calculation of its right-inverse forbidenly costy, as explained in line 104-109. The proposed blocked rearrangement convolution addresses this issue by making the receptive field non-overlapping.
>
> To be more specific, consider a 2-dimension convolution with kernel size $K \times K$, stride $S \times S$, input spatial size $H \times W$, input channel $C_i$ and output channel $C_o$. Assuming the in-place parallel matrix multiplication[1] is used. For serial method, we first substract the influence of the already-solved pixels, which takes $O(C_iK^2)$ time. Then we solve for the rest pixels, which takes $O(C_o)$ time. This substract-then-solve procedure needs to be done $HW/S^2$ times, so the overall complexity is $O(\frac{HW}{S^2}(C_iK^2+C_o))$. For parallel method implemented with the proposed blocked rearrangement, we can solve for all pixels with $O(C_o)$ time complexity. We can see that, compared with the parallel method, time complexity of serial method is forbiddenly higher and not viable for practical usages.
>
> On the other hand, non-overlapping receptive field causes visual artifact, which can be effectively removed by the proposed coupling enhancement, as is shown in Fig.4(b) and explained in line 267-274.
>
> Together, blocked rearrangement convolution with coupling enhancement provides computable right-inverse without visual artifact.
>
> **weakness-3**: We beg to differ. On the one hand, Eq. (8-9) only describe how we compute the right inverse of convolution. But $E$ and $D$ are NOT just convolutions. On the other hand, even for convolutions, a right-invertible convolution only requires $K$ to be full rank in its column, and this does NOT necessarily make $KK^+$ an identity matrix. Eq. (8) follows the convention of GEMM and writes $K$ on the right of $X$, which does not affect the order of $E$ and $D$.
>
> **weakness-4**: The idempotence is theoretically assured by right-inverse, thus cross-platform compatibility can be achieved.
>
> **weakness-5**: The insight is that near-idempotent codec does not need the right-invertibility of the encoding transform, and thus has wider choices or better model capacity.
>
> We provide the ablation study between proposed idempotent and near-idempotent frameworks, as is shown in **[rebuttal fig.3]**. It is clear that near-idempotent framework has much better first-time compression RD performance than idempotent framework. In terms of recompression RD performance, however, near-idempotent can only reach similar RD performance with much higher computation cost (8.40 GFLOPs v.s. 48.78 GFLOPs in Tab.1 and Tab.2 ).
>
> **weakness-6 & limitation**: Yes, it slows down the training speed for 2-3 times.
>
> But please note that in practical usages only inference stage is used. And inference stage is not affected because we only need the matrix, and SVD is not needed when doing inference.
>
> **weakness-7**: Thanks for the kind advice, and we will include this in the reference.
>
> [1] Randall, Keith H. (1998). Cilk: Efficient Multithreaded Computing (PDF) (Ph.D.). Massachusetts Institute of Technology. pp. 54–57
> [2] M. Lezcano-Casado. Trivializations for gradient-based optimization on manifolds. NeurIPS, 2019

---

> > ### Comment · Reviewer_H19p · 2023-08-16
> > **About weakness-2 and -4**
> >
> > I thank the authors for putting much effort into addressing most of my comments.
> >
> > (1) About the weakness-2, the authors introduce the coupling enhancement layers to increase the receptive field of the blocked rearrangement convolution. Although the blocked rearrangement convolution is parallel friendly with lower time complexity, I wonder whether the multiply-accumulate operations (blocked rearrangement convolution + coupling enhancement layers vs. ordinary convolution) may be increased in the end.
> >
> > (2)  About the weakness-4: it is widely known that different machines process floating-point arithmetic differently. Although it is argued that the right inverse is theoretically assured, how the floating-point precision issue across machines can be practically addressed in the algorithm appears to me still an issue and may impact the practicality of the proposed method.

---

> > > ### Author Response · Authors · 2023-08-17
> > > **Response to 'About weakness-2 and -4'**
> > >
> > > We thank the reviewer for thoughtful comments and feedback on our work.
> > >
> > > **(1)** The MAC of *blocked rearrangement convolution + coupling enhancement* is about 2.79 times that of *ordinary convolution*. However, actual running time is not only about MAC but also about parallelism, as is explained in the rebuttal, which makes *blocked rearrangement convolution + coupling enhancement* a lot more faster than *ordinary convolution* for computing right-inverse.
> > >
> > > **(2)** On the one hand, perfect cross-platform  consistency is hardly considered in LIC, and may prove a difficult problem itself. As is shown in [1], to perfectly avoid the floating point issue, the whole encoding-decoding process has to be carried out with only integers, and rounding after floating point calculation is not enough. Therefore, the prevalent integer flow based invertible transforms such as Helminger2021 and [2-5] may all suffer from the floating point issue.
> > >
> > > On the other hand, pure integer-integer transform can also be made surjective and thus allows for perfect right-inverse. Consider the following toy example
> > >
> > > | input | output |
> > > | -------- | -------- |
> > > | $0$ or $1$ or $2$     | $0$  |
> > > | $3$ or $4$          | $1$  |
> > > | $5$               | $2$  |
> > > | $6$               | $3$  |
> > >
> > > When we get $0$ as ouput, we can right-inverse it as $0$, $1$ or $2$ and it will still be transformed to $0$. This is a perfect right-inverse, and how to choose among $0$, $1$ or $2$ can be seen as null-space enhancement.
> > > More generally, a all-integer linear transform $Ax=y$, $s.t. A \in \mathbb{Z}^{d \times D}, x \in \mathbb{Z}^D, y \in \mathbb{Z}^d$ can be perfectly right-inversed as long as $A$ is a surjection from $\mathbb{Z}^D$ to $\mathbb{Z}^d$, though in this case much more need to be considered than mere rank of $A$ [6].
> > >
> > > To sum up, there is no contradiction between our work and cross-platform consistency issue. Thanks for pointing out a very interesting problem, which is a good future research direction. However, this issue is normally considered separately (i.e. by dedicated works like [1]) and is out of the scope of our work as well as previous works such as [2-5].
> > >
> > > [1] Ballé, Johannes, et al. "Integer networks for data compression with latent-variable models.", ICLR, 2018.
> > >
> > > [2] Hoogeboom, Emiel, et al. "Integer discrete flows and lossless compression.", NeurIPS, 2019.
> > >
> > > [3] Berg, Rianne van den, et al. "Idf++: Analyzing and improving integer discrete flows for lossless compression.", ICLR, 2020.
> > >
> > > [4] Zhang, Shifeng, et al. "ivpf: Numerical invertible volume preserving flow for efficient lossless compression", CVPR, 2021.
> > >
> > > [5] Ma, Haichuan, et al. "End-to-end optimized versatile image compression with wavelet-like transform." TPAMI, 2020
> > >
> > > [6] https://en.wikipedia.org/wiki/Diophantine_equation#System_of_linear_Diophantine_equations

---

> > > > ### Comment · Reviewer_H19p · 2023-08-22
> > > >
> > > > My comments have been addressed. I would change my rating to weak accept.

---

### Official Review · Reviewer_FGoX · 2023-07-25

**Soundness:** 3 good
**Presentation:** 3 good
**Contribution:** 3 good
**Rating:** 6
**Confidence:** 4

**Summary:**

In this work the authors address the problem of stability of codec re-compression (idempotence). In particular the paper points out the difficulty of achieving idempotence in learned image compression, and existing methods rely on invertible models such as normalizing flows which limits their performance.
The observation is that the only constraint is to have right-invertible transforms. Blocked convolutions are proposed, such that the right inverse matrix corresponding to the encoder convolution is used on the decoder side.

Besides this main point, the paper also presents additional improvements to: 1) limit the effects of the block pattern in the receptive field, 2) extend the commonly used GDN layers, and 3) address the issue with mean-shift trick quantization that doesn’t guarantee idempotence.


**Strengths:**

In general the paper is well written and clearly motivated. There are several key contributions:
+ Idea of using right inverse and its implementation with the blocked convolution and right matrix inverse.
+ Addressing all remaining details in a sound way: GDN layer, issue with the mean-shift trick in the mean-scale entropy model

It is also important to mention that has:
+ State of the art results for idempotent image compression
+ An ablation study that shows the importance of each contribution
+ Code provided along with the submission.


**Weaknesses:**

In general I think the paper is doing a great job in presenting the problem and the solution I would have minor points that authors should address in the rebuttal:

1. I think the transition from function notation to matrix notation is not smooth, and in general the transition from idempotence to right inverse could be better explained for readers less familiar with the topic of learned image compression.

2. The word “Rearrangement” together with the kernel size 5 example in line 94 brings a lot of confusion. It took me a while  to realize that the paper was proposing a new convolution and receptive fields are not overlapping which is needed to make everything work.

3. Too much space is used for the Blocked convolution while the description is not detailed enough for the rest. For example what is f(Y) exactly (eq. 11 and 12). How is it learned?

4. The extension to Near-Idempotence is a bit rushed and it’s not clear of considering the modification applied to other layers might be better?

5. In the experiments, It should always be mentioned “idempotent” or “near-idempotent” to avoid any doubt.

6. For reference, best performing non idempotent model should appear in Figure 3.a

7. For sanity check, both the idempotent version and [Helminger2021] should appear in Figure 3.b


Other details :\
l. 74: section title “Right-Invertibie” -> “Right-Invertible’\
l. 94: missing word in “if exists”?


**Questions:**

My questions are in the weaknesses (priority must be given to questions 3 and 7)

**Limitations:**

The authors discuss limitations in a sufficient manner.

---

> ### Author Rebuttal · Authors · 2023-08-09
>
> # To reviewer FGoX
>
> Thanks for your advice. We address your concerns as follows.
>
> **weakness-1**:Thanks for the kind advice. We will improve Sec.2 and Sec.3.1 in order to give a better presentation for these two points.
>
> Specifically, to better explain the transition from idempotence to right inverse for readers less familiar with LIC, in Sec.2, we will first introduce the general form for LIC, that is $y = Q \circ E(x)$ and $\hat{x}=D(y)$ ($\hat{x}$ is the reconstruction and other notations are accordingly explained). Then, we introduce the re-compression setting, which is formalized by Eq. 2. After that, we explain the idempotence requirement, which requires $E \circ D$ to be canceled. From this requirement, we give the usual solution of inverse, and finally the proposed solution of right-inverse.
>
> To make transition from function notation to matrix notation smoother, in Sec.3.1, we will first give higher level description for right-inverse of convolutions using function notation. Then, we explicitly give the shape of every variable in the function notation, for readers to get better understand of the calculation of right-inverse. Finally, we give the matrix notation, and re-emphasize the shapes so that the readers can better link matrix notation with function notation.
>
> **weakness-2**: Sorry for the unclearness. We will make this more clear by focusing on why overlapping receptive field is problematic and how do we overcome this.
>
> **weakness-3**: We will rebalance different parts and explain more on the null-space enhancement and near-idempotent setting.
>
> For $f(Y)$: $f(Y)$ is the implementation of $F$, and $F$ is the arbitrarily chosen variable in eq. 10. We choose to implement $F$ in the form of $f(Y)$ so that it is adapted to each different $Y$.
>
> Specifically, $f(\cdot)$ is learned with a common residual block, and we will include this in the experiment section.
>
> **weakness-4**: We provide additional ablation study, as follows, that tests what happens if the modification to other layers is not applied.
>
> Specifically, we test keeping the GDN layers unchanged (*w. gdn*) and keeping the convolution layers unchanged (*w. conv*). Keeping both the GDN layers and the convolution layers would reduce to baseline Balle2018.
>
> As is shown in **[rebuttal fig.1]**, keeping more layers unchanged may slightly improve first-time compression performance, but is evidently harmful to re-compression performance. The proposed near-idempotent framework has the best re-compression performance among these settings.
>
> Extra ablation study on whether changing each layer is very interesting, we are waiting for more results due to limited computational resources.
>
> **weakness-5 & weakness-6**: Thanks for your kind advices. We will modify these two points.
>
> **weakness-7**: We add idempotent codecs in Fig.3(b) for sanity check, which is shown in **[rebuttal fig.2]**. Please note that idempotent codecs are straight lines and cover each other in the figure.

---

> > ### Comment · Reviewer_FGoX · 2023-08-22
> >
> > I would like to thank the authors for the rebuttal and the additional figures. My concerns have been addressed.

---

### Author Rebuttal · Authors · 2023-08-09

To all reviewers:

please kindly reach to the pdf file below for **[rebuttal fig.x]**.

---

> ### Comment · Area_Chair_cip1 · 2023-08-13
> **Thanks for the rebuttal.**
>
> Thanks for the rebuttal. The AC and reviewers will take this into account for the final rating.

---

### Decision · Program_Chairs · 2023-09-21

**Decision:**

Accept (poster)

**Comment:**

This paper presents a novel method for idempotent image compression. It first shows that right-invertibility instead of invertibility is needed for idempotent compression. It then proposed a null space decomposition method to construct idempotent image compression network. The paper received four reviews, all of which are positive.

The paper is well written. The idea is novel and the experimental results are also convincing. The rebuttal addressed almost all questions raised by the reviewers. Please incorporate the technical details, such as the comparison with non-idemoptent  compression methods, issues with floating-point computational accuracy, etc. into the final draft.

Based on these points above, the AC made the decision to accept the paper.